# Spatio-temporal clusters and patterns of spread of dengue, chikungunya, and Zika in Colombia

**Laís Picinini Freitas**[1,2], **Mabel Carabali**[1,2], **Mengru Yuan**[1], **Gloria I. Jaramillo-Ramirez**[3], **Cesar Garcia Balaguera**[3], **Berta N. Restrepo**[4], **Kate Zinszer**[1,2]*

**1** School of Public Health, University of Montreal, Montreal, Quebec, Canada, **2** Centre de Recherche en Santé Publique, Montreal, Quebec, Canada, **3** Faculty of Medicine, Universidad Cooperativa de Colombia, Villavicencio, Colombia, **4** Instituto Colombiano de Medicina Tropical, Universidad CES, Medellín, Colombia

* kate.zinszer@umontreal.ca

**Data Availability Statement:** The data used in this study are secondary data and are publicly available. The data on dengue, chikungunya and Zika cases can be downloaded at the SIVIGILA website (http://

## Abstract

### Background

Colombia has one of the highest burdens of arboviruses in South America. The country was in a state of hyperendemicity between 2014 and 2016, with co-circulation of several *Aedes*-borne viruses, including a syndemic of dengue, chikungunya, and Zika in 2015.

### Methodology/Principal findings

We analyzed the cases of dengue, chikungunya, and Zika notified in Colombia from January 2014 to December 2018 by municipality and week. The trajectory and velocity of spread was studied using trend surface analysis, and spatio-temporal high-risk clusters for each disease in separate and for the three diseases simultaneously (multivariate) were identified using Kulldorff's scan statistics. During the study period, there were 366,628, 77,345 and 74,793 cases of dengue, chikungunya, and Zika, respectively, in Colombia. The spread patterns for chikungunya and Zika were similar, although Zika's spread was accelerated. Both chikungunya and Zika mainly spread from the regions on the Atlantic coast and the south-west to the rest of the country. We identified 21, 16, and 13 spatio-temporal clusters of dengue, chikungunya and Zika, respectively, and, from the multivariate analysis, 20 spatio-temporal clusters, among which 7 were simultaneous for the three diseases. For all disease-specific analyses and the multivariate analysis, the most-likely cluster was identified in the south-western region of Colombia, including the Valle del Cauca department.

### Conclusions/Significance

The results further our understanding of emerging *Aedes*-borne diseases in Colombia by providing useful evidence on their potential site of entry and spread trajectory within the country, and identifying spatio-temporal disease-specific and multivariate high-risk clusters of dengue, chikungunya, and Zika, information that can be used to target interventions.

portalsivigila.ins.gov.co/Paginas/Buscador.aspx).
Data can be found by disease and by year at the
SIVIGILA website, and it may be necessary to
provide an email address to download the data.
Population data can be downloaded at the National
Administrative Department of Statistics of
Colombia website (https://www.dane.gov.co/). The
processed data used for the analyses are available
at https://github.com/laispfreitas/Colombia_DZC_
satscan_velocity.

**Funding:** This work was supported by a grant from
the Canadian Institutes of Health Research (https://
cihr-irsc.gc.ca/, grant number 428107) to KZ. The
funders had no role in study design, data collection
and analysis, decision to publish, or preparation of
the manuscript.

**Competing interests:** The authors have declared
that no competing interests exist.

## Author summary

Dengue, chikungunya, and Zika are diseases transmitted to humans by the bite of infected *Aedes* mosquitoes. Between 2014 and 2016 chikungunya and Zika viruses started causing outbreaks in Colombia, one of the countries historically most affected by dengue. We used case counts of the diseases by municipality and week to study the spread trajectory of chikungunya and Zika within Colombia's territory, and to identify space-time high-risk clusters, i.e., the areas and time periods that dengue, chikungunya, and Zika were more present. Chikungunya and Zika spread similarly in Colombia, but Zika spread faster. The Atlantic coast, a famous touristic destination in the country, was likely the place of entry of chikungunya and Zika in Colombia. The south-western region was identified as a high-risk cluster for all three diseases in separate and simultaneously. This region has a favor-able climate for the *Aedes* mosquitoes and other characteristics that facilitate the diseases' transmission, such as social deprivation and high population mobility. Our results provide useful information on the locations that should be prioritized for interventions to prevent the entry of new diseases transmitted by *Aedes* and to reduce the burden of dengue, chi-kungunya and Zika where they are established.

## Introduction

Mosquito-borne diseases are thriving and expanding globally despite decades of large-scale control efforts [1]. *Aedes* mosquitoes are responsible for transmitting arboviral diseases such as dengue, chikungunya, and Zika. *Aedes aegypti* and *Aedes albopictus* are opportunist mosqui-toes adapted to urban environments for which poor quality housing and sanitation manage-ment are key determinants for the sustained propagation of arboviral diseases [2,3]. Dengue virus (DENV) is endemic in more than 100 countries with estimates ranging from 105 to 390 million infections each year [4,5]. Similarly, chikungunya virus (CHIKV) has been identified in 112 different countries, has become endemic in several countries, and is prone to explosive epidemics in areas with no prior immunity [6,7]. As with chikungunya, epidemics of dengue also occur, usually coinciding with increased *Aedes*' mosquito abundance (after a rainy season) in populations with no immunity to one of the four dengue virus types [8]. The introduction of Zika into the Americas likely occurred in late 2014, on the heels of chikungunya emergence in 2013 [9,10]. Local transmission of Zika was confirmed in 86 countries, and although Zika's Public Health Emergency of International Concern (PHEIC) is over, sporadic new cases con-tinue to be detected, indicating the potential for Zika to become endemic in previously Zika-naïve, *Aedes*-endemic countries [11–13].

 Historically, Colombia has been significantly burdened with dengue, chikungunya, and Zika infections, with the *Aedes aegypti* mosquito being widely distributed throughout the country at elevations below 2,000 meters [14,15]. The severity of outcomes of arboviral dis-eases range from asymptomatic infections, mild febrile illnesses, to severe infections that can be fatal and or produce chronic sequelae, including persistent fatigue and myalgia, debilitating joint pain, and Guillain-Barré syndrome [10,16–18]. Zika virus (ZIKV) exposure in fetuses has been causally related to microcephaly in neonates and other congenital malformations [19]. A significant portion of infected individuals remain asymptomatic and lifelong immunity can be developed for each one of the four dengue serotypes and for CHIKV and ZIKV [20–22]. There is no current antiviral treatment for arboviruses and until effective vaccines become broadly commercially available, vector control through environmental (e.g., habitat removal), chemical

(e.g., larvicide), biological (e.g., larvae eating fish), or educational campaigns remains the primary prevention strategy in most endemic settings [2,3]. Epidemiological surveillance is central to the successful control and prevention of arboviral diseases, as it provides data to monitor trends and outbreaks and identifies areas for targeted risk communication, travel advisories, and entomological response.

It is reasonable to anticipate that dengue, chikungunya, and Zika epidemiology is temporally and spatially related, given that they are transmitted by the same *Aedes* species, can co-circulate within the same region, and the presence of symptomatic infections for one virus may depend on previous or concurrent infection with one of the other viruses [23–28]. Previous work examined the spatio-temporal dependencies from 2015 to 2016 between dengue and Zika for one Colombian city and department [26]. Two other studies applied scan statistics to identify space-time clusters of arboviral diseases in Colombia, one only for dengue and chikungunya [29], and the other considered only disease-specific clusters [30]. Therefore, there have not been cluster analyses examining the co-occurrence of all three arboviruses and describing the patterns of introduction including the speed and direction of the spread of chikungunya and Zika in Colombia. We address this knowledge gap, providing meaningful insight into the shared and unique patterns of spatio-temporal disease risk in Colombia. Our study aimed to identify at-risk municipalities and time periods for dengue, chikungunya, and Zika in Colombia through disease-specific and multivariate cluster analyses and through estimating the direction and speed of chikungunya and Zika introduction in Colombia.

## Methods

This is an ecological study in which notified cases of dengue, chikungunya, and Zika from January 2014 to December 2018 for the entire country were extracted from the national surveillance system. Our units of analysis were municipality for space and week for time. Two main different methods were applied: front wave velocity analysis and scan statistics.

### Ethics statement

This study was approved by the Science and Health Research Ethics Committee (*Comité d'éthique de la recherche en sciences et en santé*—CERSES) of the University of Montreal, approval number CERSES-19-018-D. Informed consent was not required as only secondary data was used and the data were analyzed anonymously.

### Study context and data

Colombia is located at the northern tip of South America with nearly 50.4 million inhabitants. The Andes mountains dominate its topography and there are 11 distinctive geographic and climatic conditions within the country. Colombia comprises 1,123 municipalities and 33 administrative states called departments that are economically diverse and range from urban to rural to natural landscapes. Dengue became a notifiable disease in Colombia in 1978 whereas chikungunya was first detected in Colombia in July 2014 and local transmission of Zika was confirmed in the country in October 2015 [31–34].

In Colombia, surveillance is the responsibility of each department's Secretariat of Health although there is a national surveillance program administered by the National Institute of Health of Colombia that receives weekly reports from all health facilities that provide services to probable and confirmed cases of dengue, chikungunya, and Zika. The electronic platform, SIVIGILA (*Sistema Nacional de Vigilancia en Salud Pública*, http://portalsivigila.ins.gov.co/), provides publicly available aggregate data on chikungunya, dengue, and Zika cases at the municipal and departmental levels [35]. The aggregate data include probable and confirmed

cases. Probable cases are those that present clinical signs within the case definition of each disease, following the official protocols of the National Institute of Health of Colombia [31–33]. In Colombia, probable cases can be confirmed either by laboratory (based upon a positive result from antigen, antibody, or virus detection and/or isolation) or by clinical and epidemiological criteria. Complete case definitions for probable and confirmed cases for each disease are available in the S1 Appendix. Population data was obtained from the National Administrative Department of Statistics of Colombia (DANE, https://www.dane.gov.co/).

## Front wave velocity

A trend surface analysis was performed to estimate the front wave velocity for chikungunya and Zika, the two *Aedes*-diseases that emerged in Colombia during the study period. Dengue was not considered for this analysis as the disease was already endemic in the country. Trend surface analysis has been used to examine diffusion processes in two dimensions: time and space, using polynomial regression. A continuous surface is estimated with the order of the model capturing the general direction and speed of the emerging or front wave of an infectious disease [36–38].

For this analysis, we separately identified the first notification of a chikungunya and Zika case for each municipality and then used the centroids of the municipalities, calculated in meters using QGIS software [39]. The response variable was time in weeks from the first chikungunya or Zika case notified for each centroid (X and Y coordinates) in a given municipality. The continuous surface of time to notification was estimated by regressing it against the X and Y coordinates in meters. Parameters were estimated using least squares regression, and if a simple 2-D plane through the points is insufficient to model the data, high-order polynomials are often used to capture local scale trends [40]. The final models (one for each disease) were selected based on the Akaike Information Criterion (AIC), Bayesian information criterion (BIC) and expert judgment (S2 Appendix).

The rate of change was obtained by taking the partial derivatives with respect to X and Y, for the final linear models, shown below as order five polynomial (1).

$$f(t|x,y) = \beta_0 + \beta_1 X + \beta_2 Y + \beta_3 X^2 + \beta_4 Y^2 + \beta_5 X^3 + \beta_6 Y^3 + \beta_7 X^4 + \beta_8 Y^4 + \beta_9 X^5 + \beta_{10} Y^5 \\ + \beta_{11} XY \tag{1}$$

$$\partial f(t|x,y)/\partial x = \beta_1 + 2\beta_3 X + 3\beta_5 X^2 + 4\beta_7 X^3 + 5\beta_9 X^4 + \beta_{11} Y \tag{2}$$

$$\partial f(t|x,y)/\partial y = \beta_2 + 2\beta_4 Y + 3\beta_6 Y^2 + 4\beta_8 Y^3 + 5\beta_{10} Y^4 + \beta_{11} X \tag{3}$$

Eqs (2) and (3) provide expressions for a slope vector at a given location (X,Y). The vectors can be converted to express the magnitude and direction of rate of change in km per week by finding the inner product of the vector, where magnitude $\|xy\| = \sqrt{(x^2 + y^2)}$ and the direction $\theta = tan^{-1}(y/x)$. The rate we were primarily interested in was velocity (in km per week), which was obtained by inverting the final magnitude of the slope.

## Scan statistics

To identify spatio-temporal clusters we used Kulldorff's scan statistics [41]. We used a discrete Poisson model approach including the total number of cases per municipality offset by the population at-risk (municipal population). Cluster detection was performed to identify statistically significant space-time high-risk clusters of each disease separately and then performed for the three diseases simultaneously, using the multivariate scan analysis.

Kulldorff's scan statistics determine the presence of clusters using a cylinder that moves across space and time under predefined spatial and temporal scanning windows. The clusters are identified by observing a higher risk within the cylinder compared to the risk outside of the cluster. The relative risk (RR) is calculated as follows:

$$RR = \frac{\frac{c}{E[c]}}{\frac{(C-c)}{(C-E[c])}} \tag{4}$$

where $c$ and $C$ are the number of observed cases in the cylinder and the total number of observed cases in the study area, respectively, and $E[c]$ is the expected number of cases inside the cylinder, calculated as:

$$E[c] = \frac{C}{P} \times p \tag{5}$$

being $P$ the total population in the study area and $p$ the population within the cluster [41].

Clusters were ordered based on the likelihood ratio; clusters with the higher maximum likelihoods were more-likely, i.e. with stronger evidence of fitting the definition of a cluster. The likelihood function for the Poisson model is proportional to:

$$\left(\frac{c}{E[c]}\right)^c \left(\frac{C-c}{C-E[c]}\right)^{C-c} I(c > E[c]) \tag{6}$$

where $I(c>E[c])$ is set to 1 when there are more cases than expected, and 0 otherwise. For the multivariate scan analysis, the likelihood ratio of the cluster is the sum of the likelihood ratio for each disease calculated using Eq 6 [41,42].

Each space-time cluster had its own start and end dates, which were used to calculate the duration in weeks of the cluster, the number of accumulated cases observed in this period, the population within the cluster, and its relative risk.

The spatial scanning window was based on the centroid of each municipality, with the maximum size set to 150 km of radius and 20% of the total population at risk. Colombia's population is highly concentrated in a few municipalities, with only 5 municipalities accounting for nearly 30% of the country's population. Considering only the maximum population at-risk would result in clusters with very different sizes, and therefore, we also considered the maximum radius of the cylinder. To define the values, we experimented with different combinations of maximum sizes for the radius and for the population at-risk (S3 Appendix). We chose the combination that resulted in clusters that were balanced in terms of number and size: not so large as to include very distinct areas and/or low-risk municipalities, nor too small as to be too numerous and include only one municipality. The temporal scanning window was set from 2 up to 26 weeks. Clusters were restricted to having at least 100 cumulative cases over the temporal scanning window. For each model, 999 Monte Carlo simulations were performed to assess the statistical significance of observed clusters and only the clusters with p-value less than 0.05 were reported.

All analyses were performed using R (v. 4.1.2) [43] and we applied SaTScan (v. 9.6, https://www.satscan.org/) [44] using the package rsatscan (v. 0.3.9200) [45]. The front wave analysis was performed with codes based on the package outbreakvelocity (v. 0.1) [46]. The R codes and the data used in the analyses are available at https://github.com/laispfreitas/Colombia_DZC_satscan [47]. Maps were depicted using ggplot2 (v. 3.3.5) [48] and colorspace (v. 2.0) [49] packages in R, or QGIS (v. 3.22.3) [39].

## Results

During the study period, from January 2014 to December 2018, there were 366,628, 77,345 and 74,793 cases of dengue, chikungunya, and Zika, respectively, captured by the national surveillance system. Dengue cases occurred throughout this period and peaked in 2016. Chikungunya notifications rapidly increased in September 2014 and lasted for approximately a year followed by a smaller wave of cases in December 2015 to August 2016 (Fig 1). Zika notifications began increasing in October 2015, with a peak number of cases in February 2016, coinciding with a peak of dengue cases and the second wave of chikungunya. The Zika epidemic in Colombia lasted for approximately a year, following which sporadic cases were continually detected along with sporadic chikungunya cases. In 2017, there was a sharp reduction in the number of cases of the three diseases, by 84.1% compared to the previous year. Cases of dengue increased again in 2018, but chikungunya and Zika counts remained low.

The first cases of chikungunya observed in the surveillance data, at epidemiological week 23/2014, were from the municipalities of Tuluá and Barranquilla, with Tuluá being a smaller city inland in the Valle del Cauca department and over 700 kilometers away from Barranquilla, the capital city of Atlántico department, on the Atlantic northern coast. After the initial notifications, chikungunya was detected in 774 other municipalities during the first wave of cases, which lasted 14 months, following a southern pattern of dispersal from Barranquilla and a radiating pattern of dispersal from Tuluá (Fig 2A). Since the first notification of chikungunya

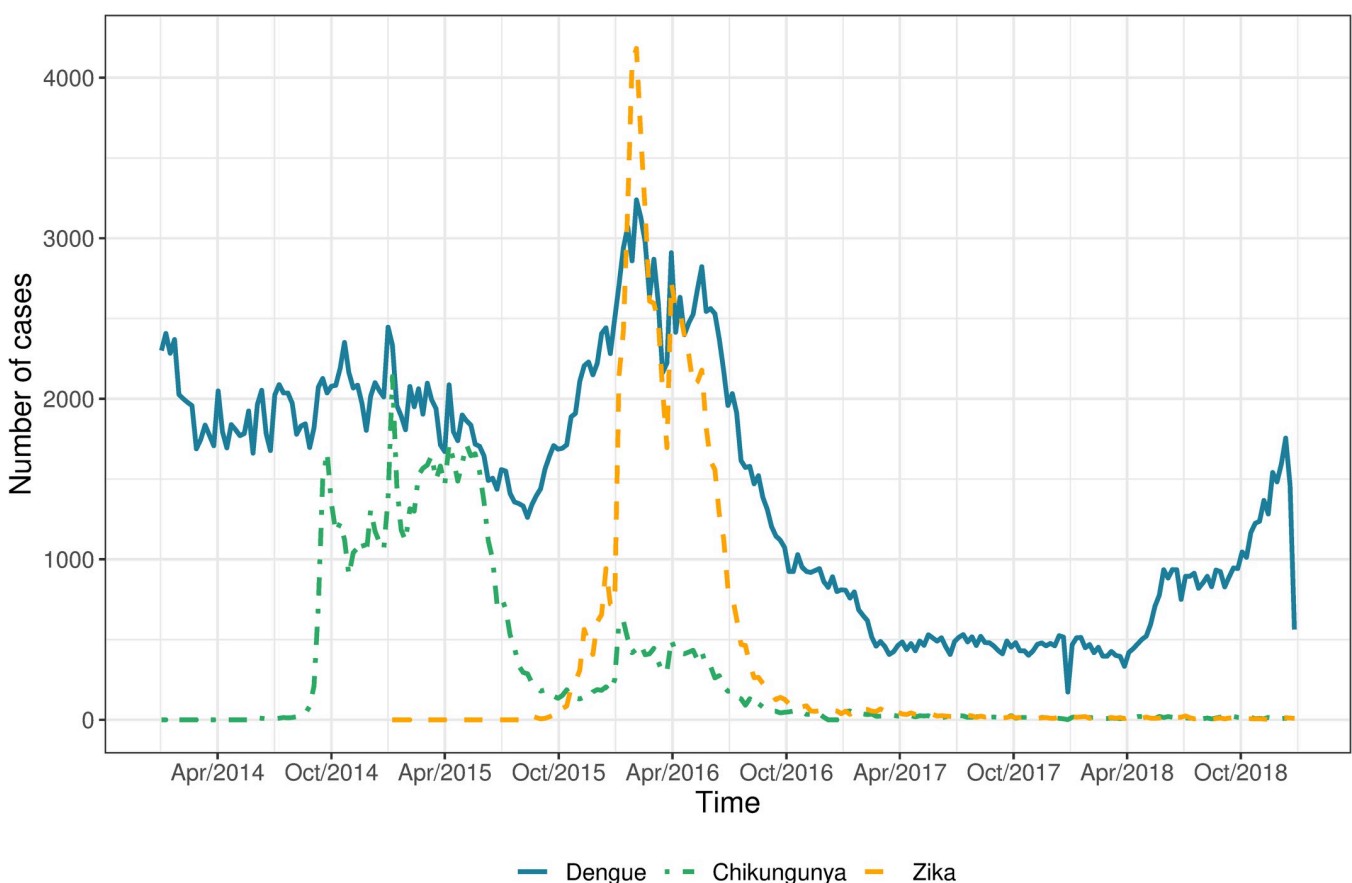

**Fig 1. Number of notified dengue, chikungunya, and Zika cases by week of first symptoms, Colombia, 2014–2018.**

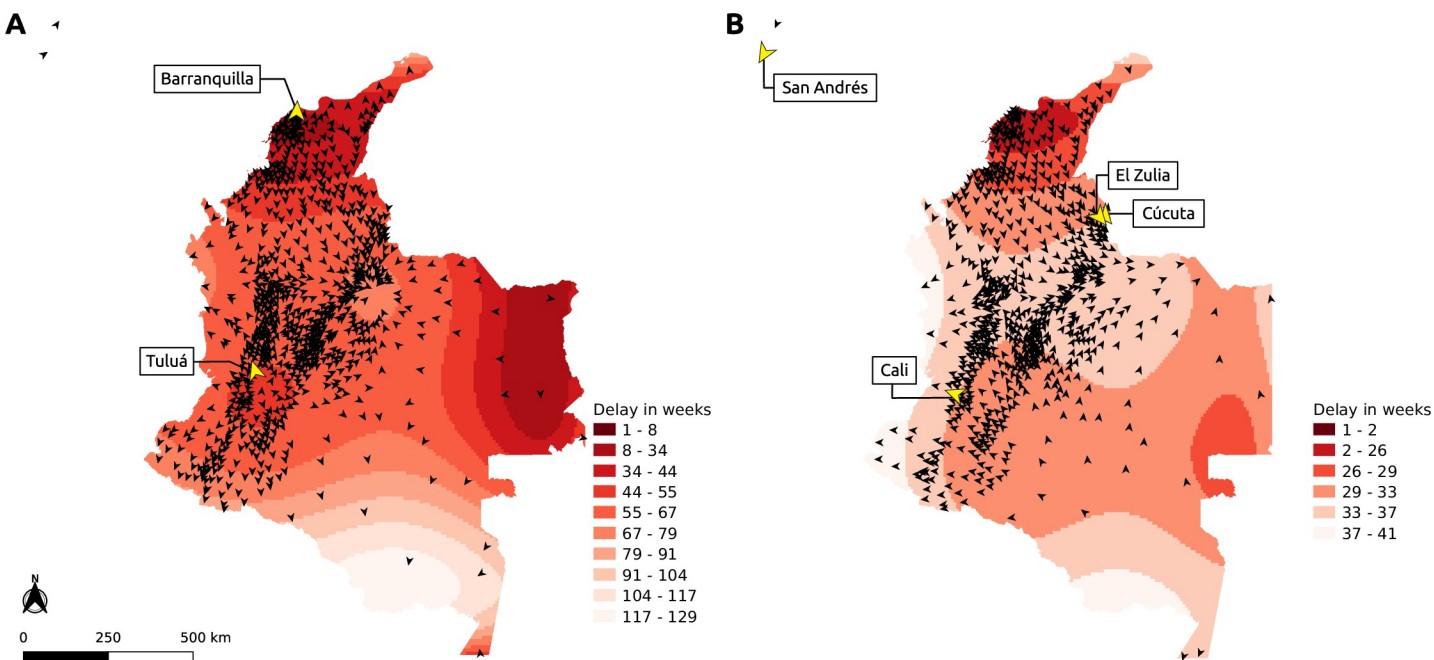

**Fig 2. Chikungunya (A) and Zika (B) spread across Colombia, 2014–2018. The angle of the arrowhead represents the direction of spread. Yellow arrowheads represent the first cases observed in the data. Base layer source: GADM (https://gadm.org/), available at** https://www.diva-gis.org/gdata.

until December 2018, it was identified in 875 different municipalities in Colombia with an average speed of 27 km/week, ranging from 1 to 397 km/week.

The first cases of Zika observed from the surveillance data, at epidemiological week 32/2015, were from four municipalities from three different departments: San Andrés (from the islands of San Adrés and Providencia, which are off the Atlantic coast of Nicaragua), Cali (capital city of Valle del Cauca department), Cúcuta and El Zulia (both from the Norte de Santander department, in the northern border with Venezuela). During the period of the Zika epidemic, it was detected in 747 municipalities following a southern pattern of dispersal from the northern Atlantic coast, a eastern pattern towards the border of Venezuela and also a northern pattern in the western part of the country (Fig 2B). Since its initial introduction and until December 2018, it was identified in 768 different municipalities with an average speed of 79 km/week, ranging from 2 to 747 km/week.

We identified 21, 16 and 13 spatio-temporal clusters of dengue, chikungunya and Zika, respectively (Tables 1 and S1–S3). When we consider the median values for each disease, Zika

**Table 1. Summary of the characteristics of the clusters for dengue, chikungunya and Zika cases, Colombia, 2014–2018.**

|  | Dengue | Chikungunya | Zika |
|---|---|---|---|
| N° of clusters | 21 | 16 | 13 |
|  | Median (IQR) |  |  |
| N° of municipalities | 5 (2–34) | 23 (9–61) | 25 (4–106) |
| Duration in weeks | 25 (21–26) | 19.5 (16–24.25) | 14 (11–21) |
| Relative risk | 5.80 (3.34–8.75) | 12.73 (7.68–21.53) | 21.43 (7.89–30.81) |
| Population inside the cluster | 203,979 (68,778–1,234,977) | 975,063 (117,505–2,018,560) | 940,273 (140,082–3,375,936) |
| N° of observed cases | 639 (204–2976) | 1207 (322–1962) | 933 (348–4663) |

IQR = Interquartile Range

case clusters included more municipalities, were shorter in duration, and had higher relative risks compared to chikungunya and dengue.

The spatial distribution of the detected clusters was similar for all diseases (Fig 3A–3C). The most likely clusters for each disease were identified in the same region, in the south-west, including the Valle del Cauca department and its capital Cali. The most likely clusters were also the largest clusters in terms of population (S1 Fig). In 2014, dengue and chikungunya clusters were detected, while in 2015 and 2016, clusters of the three diseases were detected (Figs 3D–3F and 4). In 2017, no clusters were detected and in 2018, only dengue clusters were identified (Fig 4). The first chikungunya cluster was detected in northern Colombia, on the Atlantic coast, while the Zika clusters were first identified in the islands of San Andrés and Providencia, followed by a cluster on the Atlantic coast (S2 Fig). Generally, dengue clusters lasted longer than chikungunya and Zika (Figs 3G–3I and 4) and had smaller relative risks (Fig 3J–3L). Among the clusters, higher relative risks were observed for chikungunya and Zika, particularly in the south-western part (Pacific region) of Colombia (Fig 3J–3L).

From the multivariate analysis, 20 spatio-temporal clusters were identified (S4 Table). Among those, 7 were significant for the three diseases, 5 for dengue and chikungunya, 2 for dengue and Zika, 4 clusters for dengue only and 2 for chikungunya only. The median number of municipalities forming a cluster was 14.50 (Interquartile range, IQR, 2.00–46.25). Clusters' median duration was of 16.50 weeks (IQR 14.50–25.25) and median population was of 305,307 inhabitants (IQR 5,127–8,580,330). The most likely cluster for the multivariate scan analysis was significant for dengue, chikungunya, and Zika (Fig 5A and 5B), and was also detected in the south-west of Colombia, in the Valle del Cauca department. All clusters significant for all three diseases were detected in 2015 (Fig 5C). In general, these clusters lasted longer than other clusters (Fig 5D) and presented higher relative risks for Zika compared to dengue and chikungunya (Fig 5E–5G).

## Discussion

Colombia was in a state of hyperendemicity between 2014 and 2016, with co-circulation of several arboviruses, including a syndemic of dengue, chikungunya, and Zika in 2015. Under this scenario, we explored the introduction patterns of chikungunya and Zika, and identified disease-specific and multivariate space-time clusters. This paper presents novel results on the co-occurrence of all three diseases in Colombia using multivariate cluster analysis. Our results indicate that the northern Atlantic coast was likely the place of emergence of new *Aedes*-borne diseases in Colombia, while the south-western region concentrated higher disease burden.

Overall, the geographical pattern of spread within Colombia was very similar for both chikungunya and Zika, with general southern dispersion from the Atlantic coast, although Zika spread more quickly. Our trend surface analysis and cluster evaluation identified the northern Atlantic coast, Caribbean region, for early clusters of both chikungunya and Zika, and therefore a potential portal of entry of arboviruses in the country. This region has important cargo ports on the Atlantic coast through which a good proportion of the country's imports enter. Also, there are well-known tourist sites in the region, such as Cartagena, Santa Marta and San Andrés. It is also close to the Venezuelan border, where there is an important and uncontrolled flow of people given the country's serious economic crisis. The environmental and socioeconomic conditions in this border region facilitate the transmission of infectious diseases including vector-borne diseases [50–52]. If a new pathogen was introduced in the Colombian Atlantic coastline region, its transmission and dispersion would likely be facilitated given these conditions [34]. Zika's accelerated spread is consistent with findings from another study using gravity models. The authors estimated that the spread of Zika was twice as fast as that of

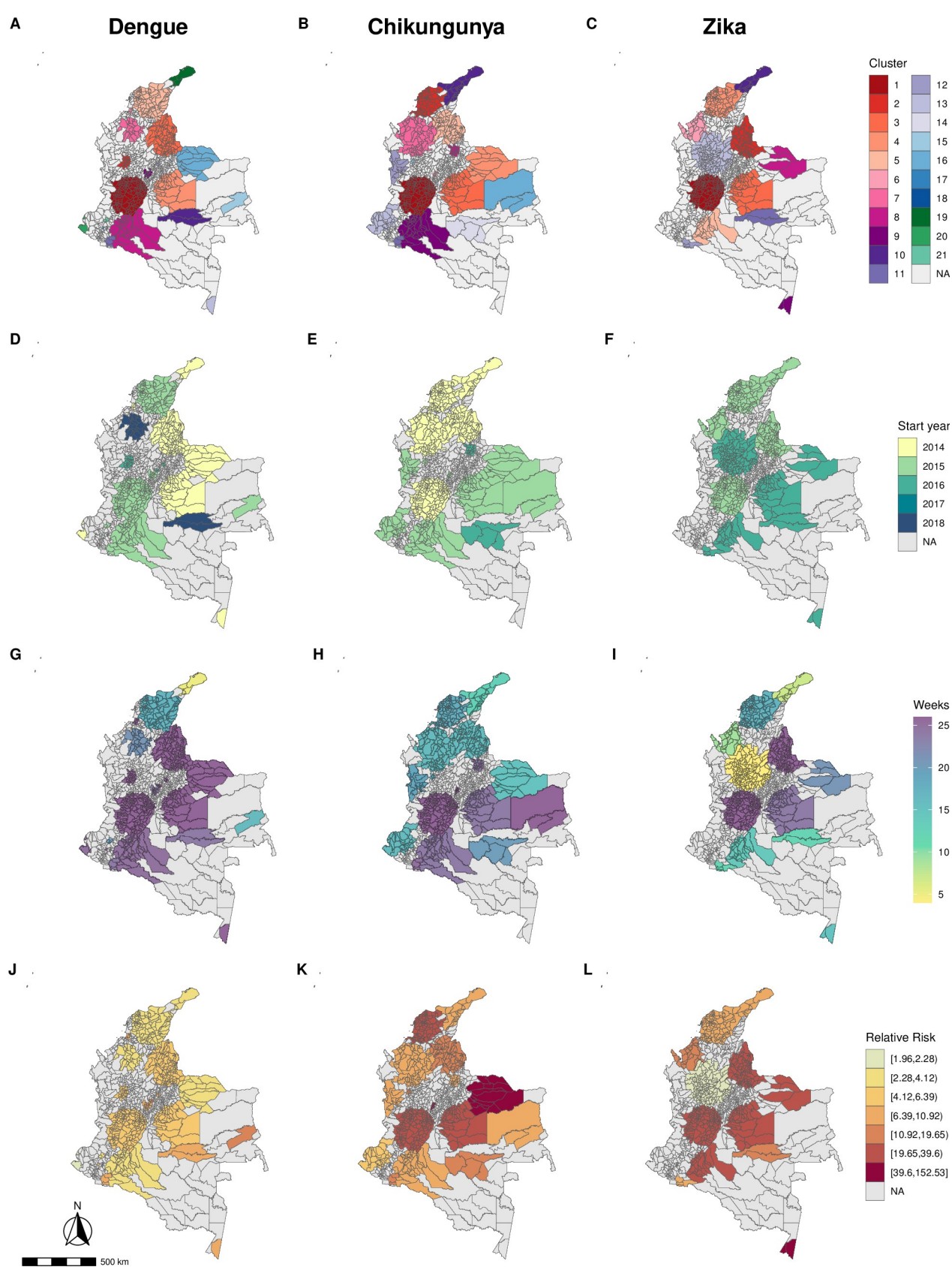

**Fig 3. Space-time clusters ranked by likelihood ratio** [*] **(A-C), year of start date (D-F), duration in weeks (G-H) and relative risk (J-L) for dengue (1st column), chikungunya (2nd column) and Zika (3rd column), Colombia, 2014–2018.** [*] The first cluster is the most likely cluster, i.e., with the maximum likelihood ratio. Base layer source: GADM (https://gadm.org/), available at https://www.diva-gis.org/gdata.

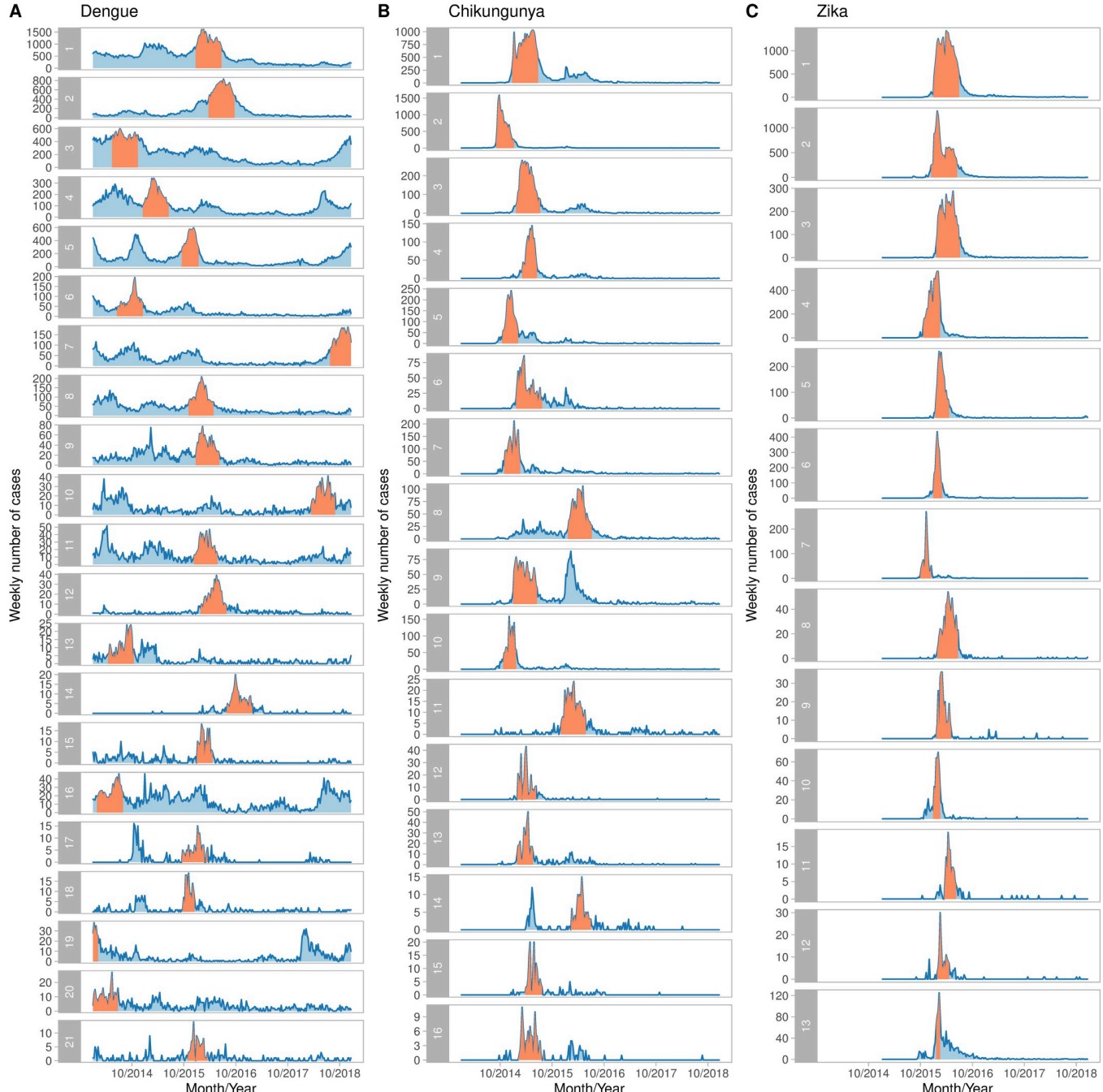

**Fig 4. Temporal distribution of cases inside each cluster of dengue (A), chikungunya (B) and Zika (C), Colombia, 2014–2018.** Orange bands represent the time period at which the cluster was detected. Clusters are ranked by likelihood ratio, being the first cluster the most likely cluster, i.e., with the maximum likelihood ratio.

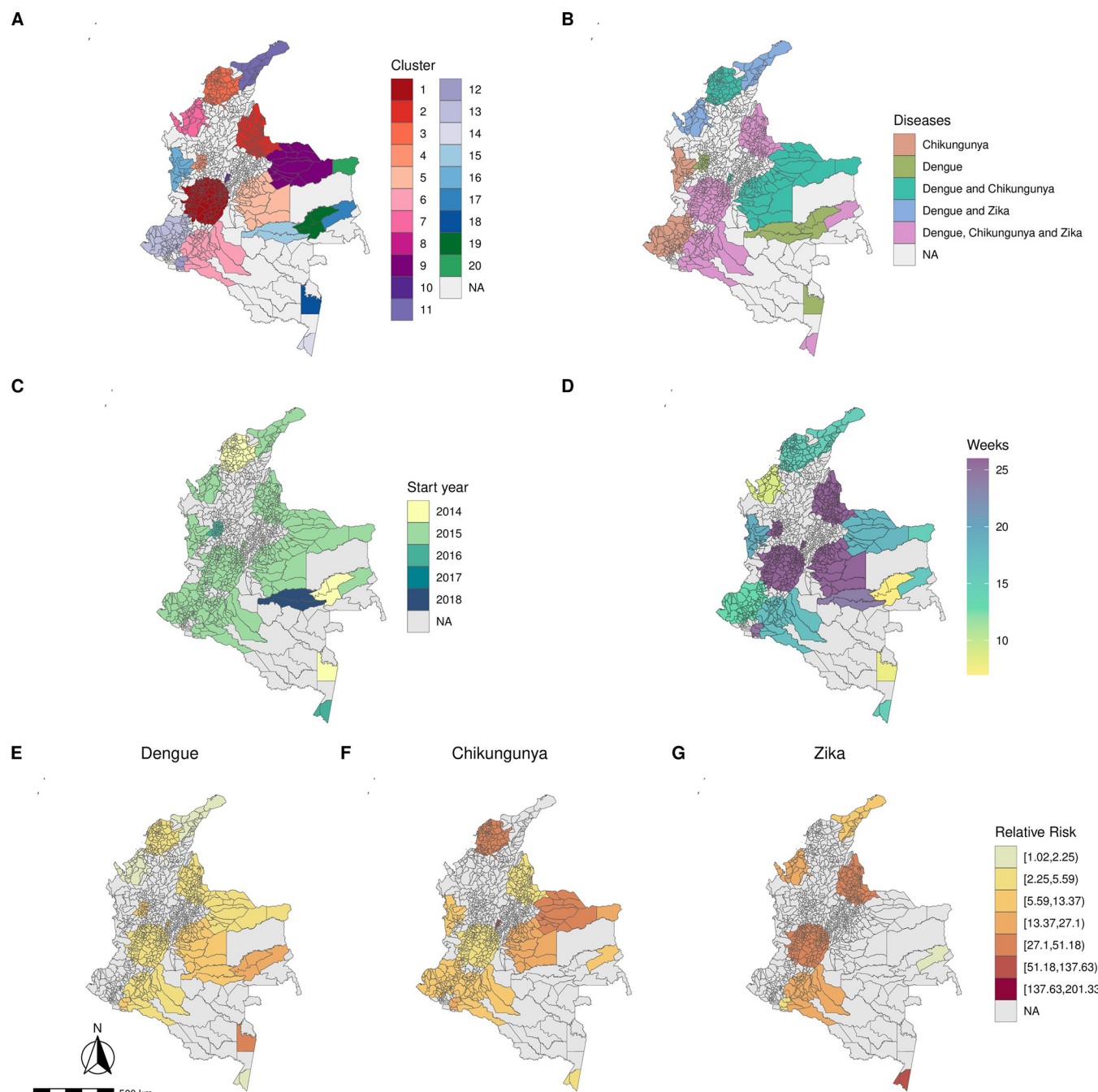

**Fig 5. Multivariate space-time clusters of dengue, chikungunya and Zika ranked by likelihood ratio** [*] **(A), diseases (B), year of start date (C), duration in weeks (D) and relative risk for each disease (E-G), Colombia, 2014–2018**. [*] The first cluster is the most likely cluster, i.e., with the maximum likelihood ratio. Base layer source: GADM (https://gadm.org/), available at https://www.diva-gis.org/gdata.

chikungunya, taking on average 15.7 weeks to invade half of the affected cities compared to 34.4 weeks for chikungunya [53]. We also observed that, compared to chikungunya, Zika's clusters were shorter and had higher relative risks, indicating a more explosive epidemic. This is possibly a result of an increased demand of people seeking medical care as a consequence of being more concerned about Zika due to the association of the virus with congenital

malformations and of neurological complications. Another possible explanation is of Zika virus being more transmissible than chikungunya, supported by evidence that *Ae. aegypti* is more efficient for transmitting the Zika virus than the chikungunya virus, even if co-infected [54,55].

For the disease-specific analyses and the multivariate analysis, the most likely cluster, i.e. the one with the strongest evidence of fitting the definition of a cluster, was consistently identified in the south-western region of Colombia, including the Valle del Cauca department and its capital Cali. This region is known for the presence of *Aedes*-borne diseases, as it has been described in other studies using the SIVIGILA data [29,30,34,56,57]. Between 2013 and 2016, Valle del Cauca was the department most affected by arboviral diseases in Colombia, accounting for 24.2% of the total disability-adjusted life years (DALYs) of the country and its capital city, Cali, contributed 13% of all arbovirus cases reported during the study period [56]. This region presents a favorable environment and suitable climate for the *Aedes* mosquitoes, with altitude less than 2000 meters above sea level, warm temperatures above 20˚C, relative humidity below 80%, and rainy seasons that are essential to the life cycle of the mosquitoes [34,58,59]. Cali, where most of the cases in the region concentrate, has a warm and dry climate, with two rainy seasons (from March to May and from September to December). The average annual total precipitation in Cali is 1483 mm, the average maximum temperature is around 30˚C, and the relative humidity of the air is on average between 70 and 76% [60]. Other factors such as environmental, cultural (e.g. water storage inside the houses) and social and material deprivation, in addition to a high population mobility due to trade with bordering countries and other cities have also favored the introduction and dissemination of arboviruses in the south-western region of Colombia [34,61–63].

Dengue, chikungunya, and Zika simultaneous clusters were identified around the cities of Cali, Ibagué and Neiva, and in western Colombia near the border with Ecuador. Cali, Ibagué and Neiva were part of the most likely clusters for all three diseases separately and simultaneously, and accounted for almost 20% of all cases of dengue, chikungunya, and Zika in the country during the study period. Cali alone contributed 13% of all cases. The western region near the border with Ecuador was previously classified as high-risk for dengue [57]. This region comprises an area with a large proportion of rural population, with limited public, educational, and health services. There is also a very active armed conflict in the region, generated by drug trafficking, further exacerbating health vulnerabilities. Therefore, the important presence of arboviruses observed in this region can be explained by its social and health inequities, in addition to a favorable climate for the presence of the *Aedes* mosquito [34].

Dengue and Zika clusters were mostly observed between the September of 2015 and the September of 2016. It is possible that during this period the transmission of *Aedes*-borne diseases was being influenced by the El Niño Southern Oscillation (ENSO). The El Niño is a climate cycle that occurs in the equatorial Pacific Ocean affecting weather patterns, and has been identified as one of the factors increasing dengue transmission in Colombia [64]. No clusters were identified in 2017, when a sharp decrease in the number of arboviral diseases reported compared to previous years was observed. This decrease is likely due to immunity development (i.e. decrease in the number of susceptible individuals to the circulating viruses), which is typical after large outbreaks [65]. In fact, we observed very small counts of chikungunya and Zika in 2017 and 2018, while dengue cases began increasing in 2018. Dengue has four circulating serotypes, and is expected to generate cyclic outbreaks every two to three years [66]. After our study period, Colombia was affected by another dengue epidemic in 2019, with numbers of cases exceeding those of 2016 [67].

The main limitations of this study are related to the quality and timeliness of the surveillance data. We used official case counts that are from a passive surveillance system, meaning

our study population included only patients who sought health care. Underreporting is a known limitation when working with surveillance data and is also an important challenge with Colombia's surveillance system [68,69]. During a syndemic of arboviral diseases that share similar symptoms such as dengue, chikungunya, and Zika, misclassification likely occurred as only a small proportion of cases are laboratory confirmed (27.2%, 4.4% and 2.7% of cases of dengue, chikungunya, and Zika, respectively, in Colombia in 2017), although differential diagnosis algorithms are used [69–72]. Earlier introductions of chikungunya may not have been captured by the surveillance system. In the case of Zika, its introduction was expected after the Brazilian epidemic, however, given the mild and generic nature of symptoms and the high proportion of asymptomatic persons, some cases may not have been captured by the surveillance system, especially in non *Aedes*-endemic areas [20]. Sporadic geographically dispersed cases were recorded in various parts of Colombia, which increased the uncertainty associated with the front wave analysis. These cases, such as those in southeastern Colombia, increased uncertainty in direction and speed estimates, which are also related to edge effects. Edge effects occurred along the boundary of the study area, which in this study were constructed by using fewer data points and are therefore less stable. One limitation of scan statistics is that the method uses circular scans to detect clusters. This could result in low-risk municipalities being considered as part of a cluster if surrounded by high risk municipalities. We counterbalanced this by restricting the clusters' size and verifying the relative risk of the municipalities forming clusters (S3 Fig). Finally, we also acknowledge that *Aedes*-borne diseases are complex systems, and that there are many other levels necessary to explain disease presence and distribution, including but not limited to environmental conditions, climate, connectivity of municipalities, human mobility, and socioeconomic and demographic characteristics.

In this study, we applied two methods to examine the emergence and spatio-temporal clusters of *Aedes*-borne diseases in Colombia. The detection of simultaneous clusters using the multivariate scan statistics analysis highlights the key hotspots areas for dengue, chikungunya, and Zika, and that should be prioritized for interventions to reduce the burden of these diseases. Additionally, these areas may also be at higher risk for other emerging *Aedes*-borne diseases. It is important that the surveillance in these locations is strengthened to be able to early detect the circulating viruses as well as unusual increase in the number of cases. Both methods are simple and provide helpful insight into the trajectory of arboviral diseases in Colombia region, which can be used to inform targeted interventions, such as enhanced surveillance activities and prevention activities. The methods have the potential to be applied for other emerging infectious diseases or variants of concern for COVID-19. It is worth mentioning that the *Aedes aegypti* mosquitoes are increasing their geographic range due to the rising global mean temperature, one of the consequences of climate change, which means regions historically not suitable for these mosquitoes are at risk of future outbreaks [73,74]. The methods described here could also be useful in these settings.

## Supporting information

**S1 Fig. Population inside clusters of dengue, chikungunya and Zika, Colombia, 2014–2018.** Base layer source: GADM (https://gadm.org/), available at https://www.diva-gis.org/gdata.
(PNG)

**S2 Fig. Week of start date of dengue, chikungunya and Zika clusters, Colombia, 2014–2018.** Base layer source: GADM (https://gadm.org/), available at https://www.diva-gis.org/gdata.
(PNG)

**S3 Fig. Relative risk for dengue, chikungunya and Zika by municipality inside a cluster, Colombia, 2014–2018.** Base layer source: GADM (https://gadm.org/), available at https://www.diva-gis.org/gdata.
(PNG)

**S1 Appendix. Case definitions for dengue, chikungunya and Zika.**
(PDF)

**S2 Appendix. Tested models for the front wave velocity analysis.**
(PDF)

**S3 Appendix. SaTScan parameters setting.**
(PDF)

**S1 Table. Space-time clusters of dengue cases, Colombia, 2014–2018.**
(PDF)

**S2 Table. Space-time clusters of chikungunya cases, Colombia, 2014–2018.**
(PDF)

**S3 Table. Space-time clusters of Zika cases, Colombia, 2015–2018.**
(PDF)

**S4 Table. Space-time clusters of dengue, chikungunya and Zika cases detected using multivariate scan statistics, Colombia, 2014–2018.**
(PDF)

## Acknowledgments

The authors would like to thank the National Institute of Health of Colombia for making the diseases' surveillance data publicly available.

## Author Contributions

**Conceptualization:** Mabel Carabali, Gloria I. Jaramillo-Ramirez, Cesar Garcia Balaguera, Berta N. Restrepo, Kate Zinszer.

**Data curation:** Mengru Yuan, Kate Zinszer.

**Formal analysis:** Laís Picinini Freitas, Mengru Yuan.

**Funding acquisition:** Kate Zinszer.

**Investigation:** Laís Picinini Freitas, Kate Zinszer.

**Methodology:** Laís Picinini Freitas, Mengru Yuan, Kate Zinszer.

**Project administration:** Kate Zinszer.

**Resources:** Kate Zinszer.

**Visualization:** Laís Picinini Freitas.

**Writing – original draft:** Laís Picinini Freitas, Mabel Carabali, Kate Zinszer.

**Writing – review & editing:** Laís Picinini Freitas, Mabel Carabali, Mengru Yuan, Gloria I. Jaramillo-Ramirez, Cesar Garcia Balaguera, Berta N. Restrepo, Kate Zinszer.

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
