## [Decision Letter · Decision Letter 0]

18 Apr 2022

Dear Dr Freitas,

Thank you very much for submitting your manuscript "Spatio-temporal clusters and patterns of spread of dengue, chikungunya, and Zika in Colombia" for consideration at PLOS Neglected Tropical Diseases. As with all papers reviewed by the journal, your manuscript was reviewed by members of the editorial board and by several independent reviewers. In light of the reviews (below this email), we would like to invite the resubmission of a significantly-revised version that takes into account the reviewers' comments. 

We cannot make any decision about publication until we have seen the revised manuscript and your response to the reviewers' comments. Your revised manuscript is also likely to be sent to reviewers for further evaluation.

Sincerely,

Alberto Novaes Ramos Jr

Associate Editor

Victor S. Santos

Deputy Editor

Reviewer's Responses to Questions

**Key Review Criteria Required for Acceptance?**

**Methods**

-Are the objectives of the study clearly articulated with a clear testable hypothesis stated?

-Is the study design appropriate to address the stated objectives?

-Is the population clearly described and appropriate for the hypothesis being tested?

-Is the sample size sufficient to ensure adequate power to address the hypothesis being tested?

-Were correct statistical analysis used to support conclusions?

-Are there concerns about ethical or regulatory requirements being met?

Reviewer #1: Freitas et al. describe the epidemiology of the triple epidemic of Zika, chikungunya, and dengue in Colombia. To do this they identify at-risk regions using spatiotemporal scan statistics, and estimate the speed of spread using trend surface analysis. The methods are clearly described and are appropriate to address the study's objectives. The study population is clearly described (although I was a little unclear on the definition of a case - see below), and I have no concerns about ethical requirements as the study used publicly available data. The specific concerns relating to the methodology that I would like to be addressed are:

- L107-113: I find the definition of a case a little confusing, please could you re-write this section. In particular it is unclear what category a positive antibody test would be in, as it is included in both definitions.

- Related to this, please could you describe how you used probable vs confirmed cases and what the breakdown of these were (i.e. the proportion in each group). 

- L128+: For the trend surface analysis, it would be useful to see the other models tested, perhaps as a supplementary table. In this table it would be useful to include the model tested, the BIC of each model, and the average and range of the estimated velocities. Including this would help the reader interpret these results by being able to compare across models.

- Please could you include a GitHub link to the code used to run the analyses.

Reviewer #2: In SatScan the authors performed a retrospective analysis of spatiotemporal clusters in Colombia. It would be interesting to see the spatial variations in the temporal trend clusters. The spatial variation in the temporal trends identifies the clusters with growth rates (or decreases) in and out of these clusters, and evaluates whether the difference between them is statistically significant.

Reviewer #3: The objectives of the study are clearly articulated and ordered in a logical manner. The study design is appropriate to address the stated objectives. The population is clearly described. I have minor concerns about the methods.

Reviewer #4: Objectives are clearly articulated, however, there is not a clear proposed hypothesis related to the transmission type of disease, seasonality, or geographical regions as initial point of the transmission. The authors can propose a hypothesis (null or alternative) for the temporal trends (seasonality or extreme events and outbreaks), or spatially related to socio-ecological drivers of the transmission (urban determinants of health, human mobility). See:

Filho, A.S.N., Murari, T.B., Ferreira, P. et al. A spatio-temporal analysis of dengue spread in a Brazilian dry climate region. Sci Rep 11, 11892 (2021). https://doi.org/10.1038/s41598-021-91306-z

Chen et al. 2019. Int J Environ Res Public Health. 2019 Jul; 16(14): 2486. Published online 2019 Jul 12. doi: 10.3390/ijerph16142486

Yes, the study design seems appropriate to explore the spatio-temporal of high-risk cluster of the diseases either separately or concurrent. 

- The population of municipalities, as unit of analysis, is described as well the number of cases by epidemiological weeks. However, what is/are the hypotheses that want to be tested? 

-Yes, municipalities jurisdiction allows a adequate analysis of the whole country.

However, the authors need to state clear the hypothesis. 

- Statistical analysis: I am not an expert in spatial statistics, so I suggest a reviewer with this expertise. However, I wonder about confounding factors such as seasonality in the different regions of Colombia. Are different patterns of rainy and hot temperatures. Considering that dengue, zika, Chikungunya are climate sensitive diseases, something to explore is how the seasonality may be a factor (or not) on the patterns of disease transmission (week number). 

-Are there concerns about ethical or regulatory requirements being met?

No, the study met all the requirements of ethics standards.

**Results**

-Does the analysis presented match the analysis plan?

-Are the results clearly and completely presented?

-Are the figures (Tables, Images) of sufficient quality for clarity?

Reviewer #1: The analysis matches the analysis plan and the results are clearly and sufficiently presented. I only have two comments:

- Could you label individually the importation locations which you mentioned in the text, so that a reader can find them.

- Figs 3 and 4 captions, I assume you rank all clusters by their likelihood (not just the first)? Perhaps better to write clusters are ranked by likelihood than the first cluster is most likely if so. Or if not, it would be better to rank them this way I think.

Reviewer #2: The results and figures are clearly and completely presented.

Reviewer #3: The results follow naturally from the analysis plan and are presented clearly. The figures are best in the .tif format included and seem of sufficient quality.

Reviewer #4: Yes, the results are linked to the objective of exploring the spatio-temporal clusters for the three diseases in Colombia. The discussion on the geographical sites for high-risk clusters is interesting and supported by the data and references. Yet, is there any additional explanation about the cities connectivity, urbanization trends, human mobility on the period of analysis?

Line 293: Could be more specific about what are the suitable climate conditions for the diseases. 

-Are the results clearly and completely presented?

The explanation of the temporal cluster analysis considering the peaks of the VBD outbreaks, would need additional discussion: potential relation with extreme years such as El Nino (2015-2016), seasonality (weeks of the year), and interannual variability (2018) compared to other studies on South America region. See:

- Rachel Lowe, Anna M Stewart-Ibarra, Desislava Petrova, Markel García-Díez, Mercy J Borbor-Cordova, Raúl Mejía, Mary Regato, Xavier Rodó. Climate services for health: predicting the evolution of the 2016 dengue season in Machala, Ecuador. The Lancet Planetary Health, Volume 1, Issue 4, July 2017, Pages e142-e151

- Petrova D et al 2020 The 2018–2019 weak El Niño: predicting the risk of a dengue outbreak in Machala, Ecuador Int. J. Climatol. 41 3813–23

The maps of figure 3 are not easy to understand, a better explanation for:

- The clusters are for all the period (2014-2018), this makes difficult to understand the dynamics of the disease’s transmission which is clearly presented in the time series. 

- Specially the relative’s risk for dengue, zika and chikungunya, individual or in co-occurrence, maybe would be better represented for various points on time (years and weeks) to visualize the temporal dynamics of the disease’s transmission.

**Conclusions**

-Are the conclusions supported by the data presented?

-Are the limitations of analysis clearly described?

-Do the authors discuss how these data can be helpful to advance our understanding of the topic under study?

-Is public health relevance addressed?

Reviewer #1: The conclusions are generally supported by the results, and the limitations are well described. I have a few comments:

- In the limitations you touch upon the effect of isolated cases, but I wonder if you could also discuss the effect of repeated introductions and how this might affect your trend surface analysis in particular.

- L268: as your model is not predictive but descriptive, please say "was likely the place of emergence" here, rather than "is likely the place of emergence".

- L279: could you briefly describe what this other study did and what they found (one sentence)

- L280: "quicker" -> "shorter"

- L266: Please remove the sentence were you say "To our knowledge, this is the first study..." as it is not informative and hard to verify.

Reviewer #2: The conclusions are supported by the data presented.

Reviewer #3: The conclusions are mostly supported by the data presented and the public health implications are discussed. I think more could be done around the limitations.

Reviewer #4: Yes, partially. The temporal dynamics need to be better explained. 

Need to expand the limitation of the study considering the following ideas:

Vector-borne diseases are complex systems which are driven by natural (climate, environment) and socio-ecological factors. The study just considered the population and number of cases of the diseases individually or in co-occurrence. However, there are many more variables that are relevant in the dynamics of the transmission. Environmental conditions, climate factors, vectors distribution, type of virus circulating, host mobility, and health and policy systems. Some of these limitations may be mentioned in this study.

-Do the authors discuss how these data can be helpful to advance our understanding of the topic under study?

Yes, with the limitation above mentioned. 

-Is public health relevance addressed?

Yes, I suggest that considering issues of global health and climate change.

**Editorial and Data Presentation Modifications?**

Reviewer #1: (No Response)

Reviewer #2: (No Response)

Reviewer #3: Line 24: (typo) the number of dengue cases reported here do not match the number reported in line 184.

Line 96: (typo) shouldn't this be 1,123 municipalities?

Line 160, 161: Missing indicator function in equation 6.

Line 184: Rewording this sentence so that there aren't 16 numbers in a row might improve readability.

Line 186-187: Split this sentence into two. One regarding dengue and one regarding CHIK.

Line 218: (typo) ", an e[a]stern pattern"

Line 268: (typo) "likely the place of emergenc[e] of new..."

Line 276-277: why is proximity to Venezuelan border a condition that would favor the introduction of new pathogens? I think additional clarification is necessary here.

Line 286: This sentence could use a bit of reworking for the sake of clarification. What does is mean for a cluster to be "most likely"? 

Lines 298-307: I think this paragraph needs to be reworked or removed. The point of the paragraph is not coming through clearly. The three cities discussed are not near the border with Ecuador, yet half of the paragraph is about proximity to Ecuador. I can't quite see how these connect.

Reviewer #4: A better explanation on the figure 3 description, and improve the visualization of the risk maps along the time.

**Summary and General Comments**

Reviewer #1: Other comment:

Abstract:

Please rewrite the third (Conclusions) section of the abstract. In particular, 1. remove the statement "To our knowledge, this is the first study...", and 2. instead of stating that this study furthers our knowledge, explain briefly how it furthers our knowledge / what knowledge it adds.

Reviewer #2: line 55: which species of Aedes is incriminated to transmit these arboviruses in Colombia?

Manuscript needs a full proofread. Text writing needs to be more fluid and easier-to-read. Maybe join similar sentences in a more organic passage without repeating the same words so much, and also create acronyms for dengue, Zika and chikungunya.

Can you clarify better the methodology applied to collect data? Were only confirmed cases considered? Or all those reported, even as suspects?

Regarding Kulldorff’s scan statistics, the authors used GINI index for optimization?

The first sentence of the discussion has already been written in the Introduction. I think the authors could start the discussion by mentioning the main findings of the study

line 284: please correct "Ae. aegypti"

In SatScan the authors performed a retrospective analysis of spatiotemporal clusters in Colombia. It would be interesting to see the spatial variations in the temporal trend clusters. The spatial variation in the temporal trends identifies the clusters with growth rates (or decreases) in and out of these clusters, and evaluates whether the difference between them is statistically significant.

The cases of these arboviruses are strictly related to the abundance of the Aedes vector. Would benefit from a short discussion about the climate (temperature, rainfall, humidity, vegetation cover) of the locations where the clusters were identified.

Is there any control method for Aedes in Colombia? e.g. some insecticide? Maybe discuss a little bit about how this could affect the distribution of cases over time

Reviewer #3: Thank you for the opportunity to review this work. This research contributes to the understanding of similarities and differences in the spatiotemporal spread of dengue, chikungunya, and Zika across Colombia. The better we can understand these dynamics, the better equipped we hopefully become to disrupt them in the future. 

Below are a handful of comments summarizing revisions or clarifications that I think would strengthen the impact of work. In particular, further clarification around the thresholds used for cluster detection *and/or* the inclusion of sensitivity analyses varying these thresholds will improve the confidence in the stability of the reported results. 

Overall, this is informative research that can be even more impactful if it more directly communicates with the findings presented in the existing literature, particularly those constructed from the same (SIVIGILA) data source.

Methods:

- line 168. A maximum radius of 150 km is set for the space-time clusters. The description of why this value was chosen is quite vague and non-informative (lines 173-175). I would like to know more about what it means to balance clusters in terms of number and size. How were decisions made about what was "too large" or "too small"? Why is a cluster comprised of a single municipality a bad thing? Previous literature on space-time clusters using municipality data have allowed clusters to be comprised of 1 municipality. Were sensitivity analyses performed for the maximum radius? This information would be useful for future decision-making in future research. I think the evidence for setting the radius could be included in the supplemental material and would be of interest to other researchers. 

- line 176. Why were clusters restricted to a minimum of 100 cumulative cases over the temporal scanning window? What effect does this have on the comparability of dengue versus CHIK and Zika clusters since there are so many fewer cases of CHIK and Zika overall? More discussion of the setting of this minimum threshold and its impact on the results would be informative. 

Results:

- Line 204. In describing the pattern of dispersal from Barranquilla, the text says that there was a southern pattern yet the arrowhead points north.

- Table 1. Related to my comment for lines 176 and 168, I wonder how these cluster results are impacted by the thresholds set on maximum distance and minimum number of cumulative cases per cluster? I recognize the need to set thresholds, but without further information on the validity of the thresholds, it would be useful to know how conclusions change as the thresholds themselves change. It is interesting that the minimum observed number of cases is highest for the disease with the lowest total case count.

Discussion:

- Line 292. Authors note that clusters of Aedes-borne diseases in the south-western region of Colombia have been identified in previous studies. Why weren’t they also identified here? Their references (29, 30) analyze the same SIVIGILA data over a similar temporal window at the same level (municipality) with, in the case of reference 29, the same scan statistic. A further discussion of how this work matches and does not match the existing knowledge generated by this data would better situate this work in the context of the literature.

Reviewer #4: The study aims to develop new knowledge examining the co-occurrence of all three arboviruses and describing the patterns of introduction including the speed and direction of the spread of chikungunya and Zika in Colombia. The authors propose to apply a cluster analysis to determine unique patterns of spatiotemporal disease risk in Colombia: through disease-specific and multivariate cluster analyses and through estimating the direction and speed of chikungunya and Zika introduction in Colombia.

Method is promising, however, the inclusion of other factors related with the complex dynamics of the VBD need to be include for future studies. Also, a better visual representation of the dynamics of the cluster may be generated. Considerations of climate variability, change and extremes would need to be included in the limitations of the study, and for future discussions of direction and speed on the region. Global and environmental changes potentially would generate new risk areas for VBD, that would need a better understanding of the dynamics for developing early warning systems for epidemics.

PLOS authors have the option to publish the peer review history of their article (what does this mean?). If published, this will include your full peer review and any attached files.

Reviewer #1: No

Reviewer #2: No

Reviewer #3: No

Reviewer #4: Yes: Mercy J. Borbor-Cordova
---

## [Decision Letter · Decision Letter 1]

12 Jul 2022

Dear Dr Freitas,

We are pleased to inform you that your manuscript 'Spatio-temporal clusters and patterns of spread of dengue, chikungunya, and Zika in Colombia' has been provisionally accepted for publication in PLOS Neglected Tropical Diseases.

Best regards,

Alberto Novaes Ramos Jr

Academic Editor

Victor Santana Santos

Section Editor

Reviewer's Responses to Questions

**Key Review Criteria Required for Acceptance?**

**Methods**

-Are the objectives of the study clearly articulated with a clear testable hypothesis stated?

-Is the study design appropriate to address the stated objectives?

-Is the population clearly described and appropriate for the hypothesis being tested?

-Is the sample size sufficient to ensure adequate power to address the hypothesis being tested?

-Were correct statistical analysis used to support conclusions?

-Are there concerns about ethical or regulatory requirements being met?

Reviewer #1: (No Response)

Reviewer #3: I would like to thank the authors for their thoughtful feedback to the previous round of comments. The objectives of the study are clearly articulated and the changes they have made have improved the clarity of the methods, results, and discussion.

**Results**

-Does the analysis presented match the analysis plan?

-Are the results clearly and completely presented?

-Are the figures (Tables, Images) of sufficient quality for clarity?

Reviewer #1: (No Response)

Reviewer #3: (No Response)

**Conclusions**

-Are the conclusions supported by the data presented?

-Are the limitations of analysis clearly described?

-Do the authors discuss how these data can be helpful to advance our understanding of the topic under study?

-Is public health relevance addressed?

Reviewer #1: (No Response)

Reviewer #3: (No Response)

**Editorial and Data Presentation Modifications?**

Reviewer #1: (No Response)

Reviewer #3: A minor point, but for the sake of completeness I would like to revisit the discussion around the indicator function in Equation 6. I acknowledge that the indicator function is left empty (i.e., "I()") in the Kulldorff SaTScanJ user guide (https://www.satscan.org/techdoc.html, "Likelihood Ratio Test"). This appears to be done strictly for user flexibility. For example, the user of the software would want to set "I(c < E[c])" to identify low rate clusters. In this work, the authors are strictly searching for high rate clusters (line 161, 177) and so the indicator function should be explicitly defined as "I(c > E[c])". This practice of explicitly defining the indicator notation is demonstrated in the two Kulldorff manuscript references included in this work (citations: 41, 42).

**Summary and General Comments**

Reviewer #1: The authors have addressed all of my prior concerns

Reviewer #3: (No Response)

PLOS authors have the option to publish the peer review history of their article (what does this mean?). If published, this will include your full peer review and any attached files.

Reviewer #1: No

Reviewer #3: No

---

## [Editor Report · Acceptance letter]

17 Aug 2022

Dear Dr Freitas,

We are delighted to inform you that your manuscript, "Spatio-temporal clusters and patterns of spread of dengue, chikungunya, and Zika in Colombia," has been formally accepted for publication in PLOS Neglected Tropical Diseases.

Best regards,

Shaden Kamhawi

co-Editor-in-Chief

Paul Brindley

co-Editor-in-Chief
